# Insulin Signaling in Intestinal Stem and Progenitor Cells as an Important Determinant of Physiological and Metabolic Traits in *Drosophila*

**DOI:** 10.3390/cells9040803

**Published:** 2020-03-26

**Authors:** Olha M. Strilbytska, Uliana V. Semaniuk, Kenneth B. Storey, Ihor S. Yurkevych, Oleh Lushchak

**Affiliations:** 1Department of Biochemistry and Biotechnology, Vasyl Stefanyk Precarpathian National University, 57 Shevchenka str., 76018 Ivano-Frankivsk, Ukraine; usemaniuk@gmail.com (U.V.S.); ihor_yurkevych@ukr.net (I.S.Y.); olushchak@yahoo.com (O.L.); 2Institute of Biochemistry, Carleton University, Ottawa, Ontario K1S 5B6, Canada; kenneth.storey@carleton.ca

**Keywords:** insulin signaling pathway, midgut, ISC, progenitor cells, lifespan, metabolism, fruit fly

## Abstract

The insulin–IGF-1 signaling (IIS) pathway is conserved throughout multicellular organisms and regulates many traits, including aging, reproduction, feeding, metabolism, stress resistance, and growth. Here, we present evidence of a survival-sustaining role for IIS in a subset of gut cells in *Drosophila melanogaster*, namely the intestinal stem cells (ISCs) and progenitor cells. Using RNAi to knockdown the insulin receptor, we found that inhibition of IIS in ISCs statistically shortened the lifespan of experimental flies compared with non-knockdown controls, and also shortened their survival under starvation or malnutrition conditions. These flies also showed decreased reproduction and feeding, and had lower amounts of glycogen and glucose in the body. In addition, increased expression was observed for the *Drosophila* transcripts for the insulin-like peptides *dilp2*, *dilp5*, and *dilp6.* This may reflect increased insulin signaling in peripheral tissues supported by up-regulation of the target of the brain insulin gene (*tobi*). In contrast, activation of IIS (via knockdown of the insulin pathway inhibitor PTEN) in intestinal stem and progenitor cells decreased fly resistance to malnutrition, potentially by affecting adipokinetic hormone signaling. Finally, *Pten* knockdown to enhance IIS also activated JAK–STAT signaling in gut tissue by up-regulation of *upd2*, *upd3*, and *soc36* genes, as well as genes encoding the EGF receptor ligands *spitz* and *vein*. These results clearly demonstrate that manipulating insulin levels may be used to modulate various fly traits, which are important determinants of organismal survival.

## 1. Introduction

The adult *Drosophila* midgut has many similarities to the mammalian intestine and is an attractive model for studying the regulation of stem cell maintenance and proliferation. Moreover, maintenance of gut integrity as an aging marker is important for fly survival and contributes to determining longevity [1]. The *Drosophila* midgut contains intestinal stem cells (ISCs) that are located on the basement membrane of the epithelium. ISCs undergo asymmetric division, where one daughter cell maintains its stemness, while the other daughter cell becomes an enteroblast (EB). Subsequently, an EB can differentiate into an enterocyte (EC) or enteroendocrine (EE) cell [2].

Regulation of ISC self-renewal, proliferation, and differentiation has a strong impact on maintaining tissue homeostasis, and in turn, organismal aging. There are several signaling pathways that control the maintenance and proliferation of ISCs. The core Notch signaling pathway interplays with other signals, including JNK, JAK–STAT, EGFR, Wg and insulin–IGF-1 signaling (IIS) pathways to regulate ISC proliferation. It has been reported that IS plays a role in the regulation of organ size by modulation of ISCs [3]. IIS can activate ISCs through two potential mechanisms: (a) enhanced division rate, and (b) switching to symmetric cell division from asymmetric division [4]. Both mechanisms lead to growth of ISCs. Moreover, IIS positively regulates ISCs proliferation during aging and regeneration [5,6]. It has been shown that the rate of ISC proliferation depends on diet and IIS activity [7]. Inhibition of IIS activity through insulin receptor (InR) loss-of-function leads to a significant decrease in the average number of pH3+ cells [8]. Choi and colleagues showed that enteroblasts non-autonomously regulate ISC proliferation in response to nutrition and IIS [6]. Enteroblasts inhibit ISC proliferation through maintenance of E-cadherin-mediated cell–cell interaction [6]. This process is controlled by the IIS pathway in enteroblast cells.

In the present work, we asked if the manipulation of IIS in a small group of gut cells may have global effects on fly physiology and metabolism and localized tissue effects. We conditionally activated or inhibited IIS in ISCs and EBs in adult flies and studied their lifespan, resistance to malnutrition, and some metabolites. In addition, gut integrity and EGFR and JAK–STAT signaling were also analyzed. We concluded that changes in IIS in both directions negatively affected fly physiology and metabolism. Unexpectedly, malnutrition conditions significantly enhanced survival of *esg/InR-RNAi* flies.

## 2. Materials and Methods

### 2.1. Fly Husbandry and Transgenic Flies

Four lines of flies (*escargot-Gal4 UAS-GFP tub-Gal80^ts^, w^1118^, UAS-InR-RNAi, UAS-Pten-RNAi*) were cultured on standard molasses medium, composed of dry yeast (5%), corn (6.1%), molasses (7.5%), nipagin (0.18%), and propionic acid (0.4%) at 18 °C for one year. The Gal4-UAS system [9] was used to drive specific interference RNAs. To downregulate the particular genes (*InR* or *Pten*) in the population of midgut stem cells, we used the temperature-sensitive *escargot^ts^* driver (*esg^ts^*), which is specific to stem cells and their immature daughter cells, enteroblasts (EBs). To reduce insulin signaling, a construct *UAS-InR-RNAi* that knocked down the insulin receptor (InR) was used. Conversely, IIS activation was achieved by knocking down the pathway inhibitor gene Pten through *UAS-Pten-RNAi* expression. Flies of the *w^1118^* line were crossed to *esg* to generate the respective control genotype *esg/+.* The efficiency of RNAi for the *InR* gene was 57% and for *Pten* was 71%, as determined by RT-PCR (Appendix A).

For all subsequent experiments (lifespan, fecundity, feeding, starvation and malnutrition resistance tests, metabolites content, and gene expression measurements), females of *Gal4* lines were mated to males of respective *UAS* lines. The resulting eggs (100 per vial) were allowed to develop at 18 °C. At this temperature, Gal80 inhibits binding of Gal4 protein to UAS. Newly enclosed flies were sexed, and selected females were kept for two additional days and shifted to 29 °C for 6 days to induce expression of the UAS constructs. Genetic crosses generated three types of transgenic fly lines: *esg/+* (control flies), *esg/InR-RNAi* (IIS inhibition), and *esg/Pten-RNAi* (IIS activation). After expression induction, the resulting flies were used for all measurements, which were conducted at 29 °C.

### 2.2. Lifespan and Fecundity

Approximately 100 females of each genotype were placed in 1.5 L demographic cages. A plastic vial filled with 5 mL of experimental food containing 5% sucrose, 5% dry yeast, 1.2% agar, and 0.18% nipagin (5%S+5%Y) was attached to the cage. Food was changed every other day, and dead flies were counted. Data were collected from two independent experiments. After 6 days of expression induction, the readout of experimental days starts initially for the survival experiments.

For each genotype, two replicates, each consisting of ten random parings with one female and two male flies, were placed into 5 mL vials with 1 mL experimental food to measure reproduction. Food was changed every day and the number of eggs laid by each individual female was recorded, as previously described [10,11]. The number of eggs was counted over a period of four days and during one day for *esc/InR-RNAi* flies after induction for 6 days.

### 2.3. Feeding

Food consumption by a single fly was measured using the CAFE assay [10]. Briefly, experimental flies were kept in a 5 µL capillary tube filled with food containing 5% yeast extract, 5% sucrose, 0.1% propionic, and 0.01% phosphoric acid. Capillaries were changed every day and the amount of food consumed was measured over a period of four days. However, flies of *esc/InR-RNAi* genotype were tested only once due to having a very short lifespan. Vials were kept in closed boxes with distilled water on the bottom to maintain high humidity for evaporation prevention. Three non-feeding vials were monitored for changes in volume to control for evaporation. Ten flies per genotype were tested.

### 2.4. Malnutrition and Starvation

For the malnutrition experiments, 15 flies were kept in 15 mL vials with 3 mL of medium containing 1% sucrose (1% S), 1% autolyzed yeast (1% AY), or 0.5% of both (0.5% S + 0.5% AY). A solution of 0.5% agarose was used for starvation assays. The vials were changed every 2 days and the number of dead flies was recorded every day for malnutrition experiments, and every 6–12–6 h (at 9:00 a.m., 3:00 p.m,. and 9:00 p.m.) for starvation experiment, until the last fly had died. Survival of 45–75 flies per genotype was tested.

### 2.5. Metabolites

Glucose and trehalose in hemolymph, as well as glucose, trehalose, and glycogen in fly bodies, was measured as described previously [12]. Briefly, preweighed flies were decapitated and centrifuged to extract hemolymph (3000× *g*, 5 min). Fly bodies were homogenized in 50 mM sodium phosphate buffer (pH 7.0), centrifuged, and used for determination of glucose and glycogen levels. Measurements were performed using a glucose assay kit (Liquick Cor-Glucose diagnostic kit, Cormay, Poland, Cat. No. 2-203). The glycogen was converted into glucose by incubation with amyloglucosidase from *Aspergillus niger* (25 °C, 4 h) and then glucose was measured as above. The levels of body glucose and glycogen were displayed as milligram per gram of wet weight (mg/gww). For triglyceride (TAG) determination, flies were weighed, homogenized in 200 mM phosphate buffered saline containing 0.05% Triton X100 (PBST), boiled, and centrifuged (13,000× *g*, 10 min) [13]. Resulting supernatants were used for TAG assay with Liquick Cor-TG diagnostic kit (Cormay, Poland). Flies of all genotypes were tested in four independent replicates.

### 2.6. Analysis of Gut Integrity

To determine intestinal integrity, we examined flies that consumed food supplemented with non-absorbable blue food dye E133 [1]. “Smurf” flies were defined by visible blue food dye seen throughout the body, which suggests disruption of gut integrity. In total, 150 females of each genotype were placed into plastic vials with food supplemented with 2.5% E133, and after 12 h, the number of “smurf” flies was counted. The experiment was repeated every 5 days for up to 30 days (on day 1, 5, 10, 15, 20, 25, and 30). As the flies of *esg/InR-RNAi* genotype died quickly, fast gut integrity was only measured once on experimental day 1.

### 2.7. Gene Expression

Total RNA from heads, whole flies, or dissected guts was extracted with RNeasy Plus Mini Kit (Qiagen, Hilden, Germany) and converted into cDNA with QuantiTect Reverse Transcription Kit (Qiagen). Expression of genes of interest was measured using an ABI Prism 7000 instrument (Applied Biosystems, Foster City, CA, USA), a SensiFAST SYBR Hi-ROX Kit, and a QuantiTect SYBR Green PCR Kit (Qiagen) under conditions recommended by the manufacturers. Each analytical and standard reaction was performed in three technical replicates. The levels of transcripts were measured using previously published primer pairs, shown in Appendix A [14,15]. The Ct method was used with *rp49* (ribosomal protein 49) as the reference gene for heads and whole flies, while *crq* (croquemort) was the reference gene to assay samples from the gut.

### 2.8. Statistical Procedures

To assess statistically significant changes in lifespan and survival in response to experimentally induced stresses, survival curves were constructed from the starvation, malnutrition, and oxidative stress resistance assays, and analyzed with the log-rank (Mantel-Cox) test (Prism GraphPad, version 6). Differences between groups were analyzed using ANOVA, followed by Newman–Keulspost post hoc test (Prism GraphPad, version 6).

## 3. Results

### 3.1. Lifespan and Stress Resistance

The role of the IIS pathway in lifespan regulation is conserved across eukaryotic organisms from nematodes to mammals. The mean lifespan of control flies *esg/+* was approximately 25 days (range 25–26 days). Inhibition of IIS signaling in ISCs and EBs (*esc*-cells) due to *InR-RNAi* expression accelerated the mortality of flies by the second experimental day (log-rank, *p* < 0.0001; χ^2^ = 144) (Figure 1A). No significant difference was observed in survival rate between flies with activated IIS in *esg*-cells (*esg/Pten-RNAi*) and *esg/+* control flies.

Nutrient availability is a key determinant of organismal growth and survival. The effects of malnutrition on fly survival were also evaluated in flies fed with restricted diets (i.e., either 1% sucrose (S), 1% autolyzed yeast (AY), or 0.5% sucrose + 0.5% AY) (Figure 1). Interestingly, malnutrition prolonged the lifespan of the flies expressing *InR-RNAi* when compared to the control diet. Although the flies with *InR-RNAi* expression in *esg*-cells lived only 2 days on the control diet or during starvation (Figure 1E), diet conditions of 1% sucrose, 1% AY, or 0.5% of both components increased mean lifespan to 6, 7, or 9 days, respectively (Figure 1B–D). However, resistance of the modified flies, either *InR-RNAi* or *Pten-RNAi* genotypes, was in all cases significantly lower as compared to the *esg/+* control phenotype. Flies that express *Pten-RNAi* in *esg-*cells exhibited decreased malnutrition resistance on a diet of 1% sucrose (log-rank, *p* = 0.02; χ^2^ = 5) (Figure 1B) and 1% AY (8% and 33%, respectively) (log-rank, *p* = 0.01; χ^2^ = 11) (Figure 1C). However, a balanced low-calorie diet (0.5% sucrose and 0.5% AY) had no significant impact on survival of *esg/Pten-RNAi* as compared to *esg/+* flies and significantly reduced survival of *esg/InR-RNAi* flies by 56% (Figure 1D).

We also tested the survival of transgenic flies under a condition of complete starvation (0.5% agarose diet). More precisely, these flies also lived for only 2 days under complete starvation (Figure 1E). These *esg/InR-RNAi* flies exhibited a significant decrease in resistance to complete starvation by 60% compared to *esg/+* control flies (log-rank, *p* < 0.0001; χ^2^ = 90) (Figure 1E; Appendix A). Moreover, we observed a significant decrease in survival (8%) under starvation conditions in flies expressing *Pten-RNAi* in *esg*-cells (log-rank, *p* = 0.001; χ^2^ = 10) (Figure 1E).

### 3.2. Feeding and Fecundity

We investigated the consequences of IIS modulation in stem and progenitor cells on feeding and fecundity rates, which has a profound effect on *Drosophila* lifespan. Control flies (*esg/+*) consumed an average of about 1.3 µL of food and laid approximately 14 eggs daily. IIS inhibition in midgut stem and progenitor cells decreased food consumption by 52% (Figure 2A; *p* < 0.05) and fecundity by 74% (Figure 2B; *p* < 0.05) as compared to control. IIS pathway activation through *Pten-RNAi* expression in *esg*-cells resulted in 43% higher food intake as compared to *esg/+* flies (Figure 2A; *p* < 0.05). Moreover, these flies showed a 23% increase in the daily production of eggs (Figure 2B; *p* < 0.05).

### 3.3. Metabolism

Insulin signaling plays a key role in maintaining metabolic homeostasis. To determine whether modulation of the IIS pathway in stem and progenitor cells has global effects on *Drosophila* metabolism, we measured glucose and glycogen levels in the whole body of flies. We found that *InR* knockdown in ISCs decreased the level of whole-body glucose by 20% versus controls (Figure 3A; *p* < 0.05). Interestingly, trehalose levels within hemolymph or in the body were not affected by IIS modulation in *esg*-cells (not shown). Examination of stored fuel reserves revealed that *InR-RNAi*-expressing flies in *esg*-cells contained significantly lower glycogen levels, in this case a 35% decrease as compared to controls (Figure 3B; *p* < 0.05). However, there were no effects of IIS modulation in ISCs on TAG storage (Appendix A).

### 3.4. Expression of DILP Genes

*Drosophila* insulin-like peptides (DILPs) regulate glucose metabolism [16], however a clear definition of the functions of the various DILPs has not yet been achieved. The manipulations of IIS in *esg*-cells in this study affected the steady-state transcript levels of *dilp2*, *dilp5*, and *dilp6* (Figure 4). Significantly higher relative expression of *dilp2* in fly heads was found when IIS was inhibited in ISCs (by 77%) (Figure 4A; *p* < 0.05). A similar result was observed for *dilp5* transcripts in fly heads, with a 50% increase in *dilp5* expression levels in *InR-RNAi*-expressing *esg*-cells (Figure 4C; *p* < 0.05). Interestingly, both activation and inhibition of IIS in *esg*-cells led to higher whole-body *dilp6* transcript levels, reaching nearly 2-fold increases (Figure 4D; *p* < 0.05). Expression of *InR-RNAi* and *Pten-RNAi* in *esg-*cells did not affect the relative expression of *dilp3* (Figure 4B).

### 3.5. Expression of Genes Related To Glucagon-Like Signaling and Metabolism

The adipokinetic hormone (AKH) plays an important role in *Drosophila* metabolism and serves as a determinant of carbohydrate and lipid levels [17]. We found a 2-fold increase of *akh* transcript levels in whole *Drosophila* body when IIS was activated in ISCs of *Pten-RNAi* flies (Figure 4E; *p* < 0.05). The target of brain insulin *tobi* is regulated by both insulin and glucagon signaling, and its increased mRNA level might cause a decrease in glycogen content [18]. *InR-RNAi* expression in ISCs led to a 3-fold increase in *tobi* transcript levels in whole *Drosophila* bodies (Figure 4F; *p* < 0.05). We also observed a 1.5-fold increase in *tobi* transcript levels when IIS was activated in *esg*-cells expressing *Pten-RNAi* (Figure 4F; *p* < 0.05).

*Drosophila* PEPCK is involved in gluconeogenesis and glycerogenesis [19]. Both genetic manipulations in *esg-*cells had no impact on *pepck* transcript levels in whole *Drosophila* bodies (Figure 4G). We also did not observe any significant changes in *4ebp* transcripts when IIS was modulated in *esg*-cells (Figure 4H).

### 3.6. Gut Tissue-Specific Effects

Since stem and progenitor cells are important for maintenance of gut tissue homeostasis, we evaluated gut integrity and JAK–STAT and EGFR signaling pathways, which are involved in regulation of ISC proliferation and EB differentiation [8,20]. Measurements of gut integrity with the smurf assay (dye diffusion to other tissues as a result of gut damage) [1] showed that perturbation of IIS did not affect tissue integrity. In all cases, incidence of “smurf” flies was lower than 7% of the total flies (Appendix A).

We found that transcript levels of the cytokine-inducing gene, *upd2*, were 4-fold higher in the gut of flies expressing *Pten-RNAi* in *esg*-cells (Figure 5A; *p* < 0.05). Similarly, transcript levels of *upd3* also increased 3-fold in the gut of these flies (Figure 5B; *p* < 0.05).We also measured the transcript level of *soc36*, which is a target for JAK–STAT signaling, and found a 50% increase in *soc36* transcript levels in the gut of *Pten-RNAi*-expressing flies (Figure 5C; *p* < 0.05). No change in transcript levels of these three genes occurred in the gut of flies expressing *InR-RNAi* in *esg*-cells. Hence, *upd2, upd3*, and *soc36* displayed similar trends in relative expression within the groups of transgenic *esg* flies.

Gene transcripts to EGFR ligands, namely *spi* (Spitz), *krn* (Keren), and *vn* (Vein), were also measured in the fly gut. Our data demonstrated significant increases in *spi* and *vn* transcript levels (~2.4 fold) when IIS was activated in *esg*-cells (Figure 5D, F; *p* < 0.05). Moreover, we observed 2-fold higher relative *vn* expression in flies expressing *InR-RNAi* in *esg*-cells (Figure 5F; *p* < 0.05). However, modulation of IIS activity did not affect *krn* transcript levels (Figure 5E).

## 4. Discussion

The IIS pathway regulates diverse physiological processes in multicellular organisms, including growth, reproduction, metabolism, and longevity [16]. Reduced expression of the insulin receptor [21] and *Pten* overexpression [22] both have been shown to extend *Drosophila* lifespan. Similar effects were observed when insulin-producing cells were ablated [16]. These results suggest an important role of IIS in aging. In addition, loss-of-function genetic studies in *Caenorhabditis elegans* have shown that the nematode insulin receptor, daf-2, prolonged lifespan [23]. In mice, deletion of *InR* in white adipose tissue also increased longevity [24]. We used *InR-RNAi* expression to inhibit IIS in flies and *Pten-RNAi* expression to functionally activate IIS by decreasing PTEN phosphatase mRNA levels in stem and progenitor cells of fruit fly gut (Figure 6).

Table 1 summarizes all results of this study in order to compare activation versus inhibition of IIS. Our study revealed that *InR* knockdown in stem and progenitor cells caused detrimental effects on fly survival, supporting the hypothesis that IIS plays an important role in the normal functions of stem cells. Since IIS plays different roles in different tissues, the resulting phenotypes from modulated IIS activity can be different. We propose that affecting the insulin receptor in stem and progenitor cells may have whole organismal effects. When the insulin receptor is knocked down, stem cells signal to other cell types in the gut, such as enterocytes and enteroendocrine cells. Signaling to these cells may reduce nutrient uptake in the gut, suggesting starvation. Inhibition of stem cell activity may also induce loss of tissue integrity [8]. Importantly, manipulations of IIS in both cell types did not induce damage to gut tissue. The smurf assay showed no change in the percentage of flies with damaged (leaky) guts in control versus experimental groups (Appendix A). An absence of effects on gut integrity when insulin signaling was disrupted might be explained by changes in JAK–STAT and EGFR signaling. The unpaired (upd) 2 and 3 proteins activate the JAK–STAT pathway activity in ISCs [20]. To evaluate the mechanisms involved in maintenance of gut homeostasis, we tested the possible involvement of the JAK–STAT and EGFR pathways. The two pathways regulate ISC proliferation [21] (Figure 6). *Upd2* and *upd3* expression in enterocytes, enteroblasts, or stem cells in the *Drosophila* gut lead to enhanced ISC proliferation, which in turn causes hyperplasia [21]. EGFR signaling is necessary for ISC proliferation [22]. We tested the relative expression of genes encoding ligands for JAK–STAT (*upd2, upd3*) and its target gene (*soc36*), as well as ligands for EGF receptor (*spi, krn, vn*) signaling pathways. Our data demonstrate a highly significant increase in *upd2* and *upd3* transcripts when IIS was activated as a result of PTEN suppression, indicating that IIS can activate JAK–STAT signaling in the gut. Moreover, increased relative expression of *soc36* in *Pten-RNAi* expressing flies confirms the interdependence between IIS and JAK–STAT. The EGFR pathway activity may be activated in the adult intestine by the ligands Vein (Vn), Spitz (Spi), and Keren (Krn), which are able to bind to the EGF receptor [25]. Keren is produced by ECs [26], but neither inhibition nor activation of IIS in ISCs affected *krn* expression. Furthermore, IIS activation increased *spi*, which was detected in EBs, as well as *vn* expression, which was detected in both ECs and visceral muscle [26]. These data suggest that manipulations of insulin signaling in stem and progenitor cells affect other cell types and involve signaling specifically linked to EGFR and JAK–STAT.

Manipulations with IIS in midgut stem and progenitor cells affected fly lifespan, as well as sensitivity to starvation and malnutrition. We observed rapid mortality within 2 days in flies with *InR* knockdown in stem and progenitor cells. Our results are consistent with the study of Biteau and colleagues, which showed an impact of IIS on stem and progenitor cells, shortening lifespan when IIS was inhibited in both cell types [8]. However, by contrast, our data demonstrated greatly shortened lifespan only when IIS was inhibited in ISCs. Moreover, these transgenic flies were short-lived under complete starvation. These effects may be explained by a reduced gut size, which has been previously shown to be due to a decrease in insulin signaling [3]. Reduced gut size also explains the decrease in feeding by more than 50%, as well as the reduction in body fuel reserves of glucose and glycogen. In addition, these flies showed increased transcription of *dilp* 2, 5, and 6 with increased insulin signaling to peripheral tissues, as indicated by the induction of *tobi*. We also observed an increase in *dilp6* transcript levels in flies with activated IIS in *esg-*cells. Mutations of *dilp6* significantly reduced fly fecundity [27], suggesting that up-regulated *dilp6* can increase fly fecundity [28]. In addition, increased food uptake may contribute to increased fecundity by supplying more nutrients for egg production. This may be because the *esg* driver might be expressed in the other *Drosophila* tissues. Indeed, according to FlyAtlas, the *esg* driver is also expressed in fly testis. Consequently, there are some potential contributions from other cells and tissues to the systemic assays performed. One more contributor to these results may be the up-regulation of the PEPCK transcript, an important enzyme for maintaining the balance between catabolic and anabolic processes [29]. Induction of *pepck* transcription could be stimulated by the strong induction of AKH signaling as a result of the 2-fold increase in *akh* transcript levels.

Interestingly, malnutrition partially rescued the longevity phenotype in flies with decreased IIS in stem and progenitor cells. Under a normal diet, the mean lifespan of flies with inhibited IIS was about two days, however diet with only 1% sucrose enabled flies to live up to 6 days. In addition, flies fed the low protein diet (1% AY) lived 7 days and those fed low-carbohydrate low-protein food (0.5% sucrose and 0.5% AY) lived up to 9 days (Figure 7). These results clearly show that IIS plays a key role in mediating the effect of dietary regimens on overall physiology. Different mechanisms and pathways are involved in detecting the intracellular and extracellular levels of macronutrients, including the TOR signaling pathway, which has a significant impact on lifespan [30,31]. Moreover, we have previously demonstrated that TOR signaling inhibition, which acts in a strong interaction with IIS, in *Drosophila* ISCs-EBs can have important deleterious consequences for regenerative capacity, which in turn decreases resistance to malnutrition [32]. However, our results showed that TOR signaling activation through *rheb* expression in *Drosophila* ISCs and EBs increases life expectancy under conditions of malnutrition (1S, 1Y), which suggested that activation of TOR signaling can offset malnutrition as a stressful agent [15].

Contrasting phenotypes for IIS activation and inhibition in esg cells were observed for fecundity and feeding. Both parameters were reduced as a result of decreased IIS, but were increased when IIS was activated. However, in the case of decreased malnutrition (1% sucrose) resistance and enhanced *dilp6* transcript levels, we found the same effects under either IIS activation or inhibition. Many parameters, such as lifespan, body glucose, and glycogen levels, transcripts of *dilp2, dilp5*, and *tobi* were affected only by a reduction of IIS. However, gut-specific increases in mRNA transcript levels of *upd2, upd3, soc36, spi*, and *krn* were seen only in flies with activated IIS in esg cells (Table 1). Thus, current data provide evidence about the significance of cooperation between multiple signaling pathways, including JAK–STAT, downstream of EGFR and insulin signaling for regulation of stem cell maintenance and activity.

## 5. Conclusions

Altogether, the work described here reveals the importance of insulin signaling in midgut stem and progenitor cells. Activation or inhibition of the insulin signaling pathway influenced fly physiology and metabolism, and affected transcription of genes involved in the regulation of metabolism, gut homeostasis, and survival, both in the whole body and in the gut itself. These results clearly demonstrate the possibility to modulate various fly traits through interactions with insulin signaling in the gut stem and progenitor cells.

## Figures and Tables

**Figure 1 cells-09-00803-f001:**
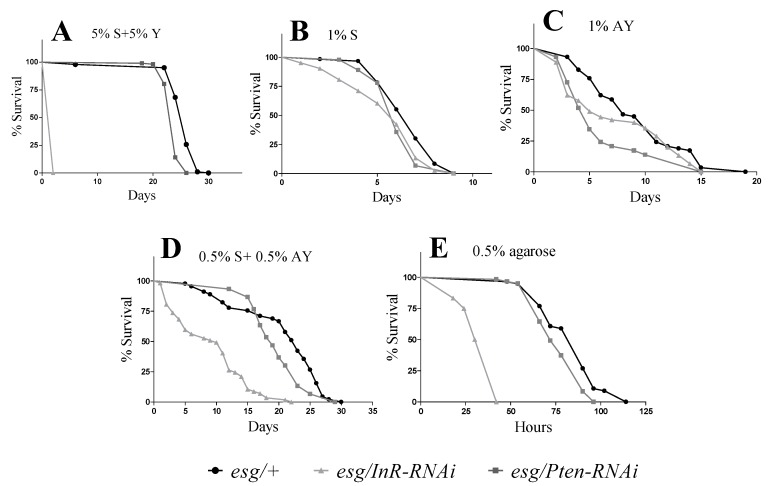
Survival curves for the experimental flies with insulin–IGF-1 signaling (IIS) manipulated in stem and progenitor cells under various dietary conditions: **A**—standard medium (5%S + 5%Y); **B**—low carbohydrate diet (1%S); **C**—low protein diet (1%AY); **D**—balanced malnutrition diet (0.5%S + 0.5%AY); **E**—complete starvation (0.5% agarose). The readout of the experimental days starts initially after six days of expression induction at 29 °C. Each curve represents the percentage of individuals alive. The genotypes were compared using log-rank test (see Appendix A for complete statistics).

**Figure 2 cells-09-00803-f002:**
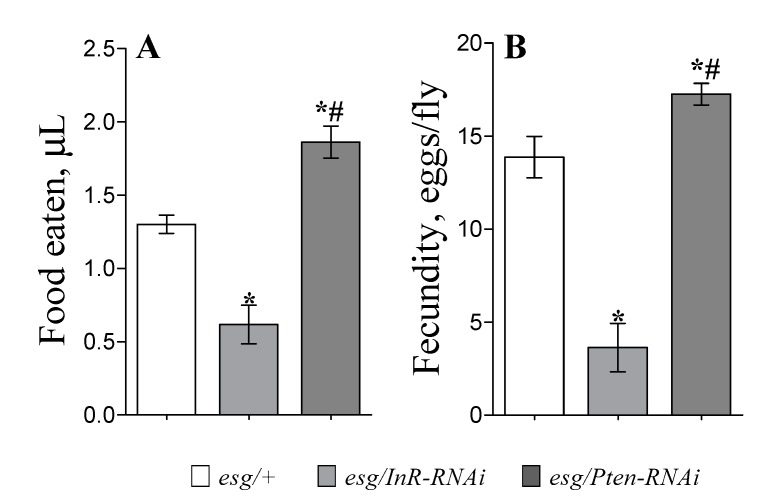
Daily food consumption (**A**) and number of eggs laid daily per female (**B**) for flies with IIS, either activated or inhibited in *esg-*cells. Results are presented as mean ± SEM, with 10–20 flies tested per genotype. The experiments started after six days of expression induction at 29 °C and were performed over four days. Due to their reduced lifespans, *esc/InR-RNAi* genotype flies were tested ones; thus, bars present results of single-day measurements. Group comparisons were performed using ANOVA followed by Newman–Keuls post hoc test. Asterisks indicate significant difference from control flies (*esg/+*) (*p* < 0.05); hash signs show significant difference from flies with inhibited IIS (*esg/InR-RNAi*).

**Figure 3 cells-09-00803-f003:**
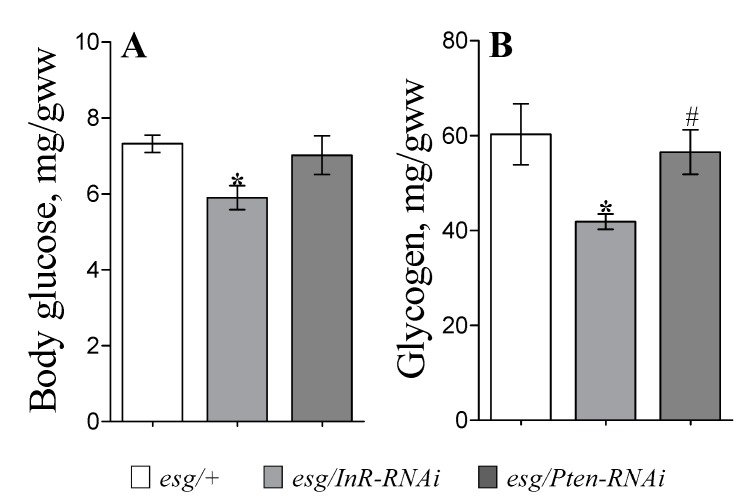
Body glucose contents (**A**) and glycogen amounts (**B**) in flies with IIS modulated in stem and progenitor cells. Flies were used after six days of expression induction by fly transfer to 29 °C. Results represent the mean ± SEM of 4 or 5 replicates per genotype. Group comparisons were performed using ANOVA followed by Newman–Keuls post hoc test. Asterisks indicate significant difference from the control flies (*esg/+*) (*p* < 0.05); hash sign shows significant difference from flies with inhibited IIS (*esg*/*InR-RNAi*).

**Figure 4 cells-09-00803-f004:**
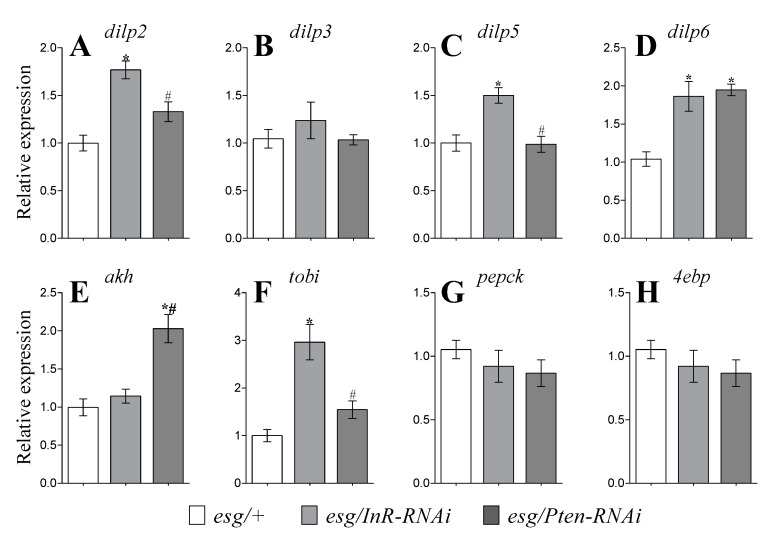
The mRNA levels for *dilp2* (**A**), *dilp3* (**B**), and *dilp5* (**C**) in fly heads and *dilp6* (**D**) from whole fly bodies and transcript levels of genes related to glucagon-like signaling and metabolism: *akh* (**E**), *tobi* (**F**), *pepck* (**G**), and *4ebp* (**H**) from whole fly bodies. Parameters were measured in flies kept at 29 °C for six days for RNAi expression induction. Data are mean values for 4 independent measurements (± SEM). Group comparisons were performed using ANOVA followed by Newman–Keuls post hoc test. Asterisks indicate significant difference from the control flies (*esg/+*) (*p* < 0.05); hash signs show significant difference from flies with inhibited IIS (*esg/InR-RNAi*).

**Figure 5 cells-09-00803-f005:**
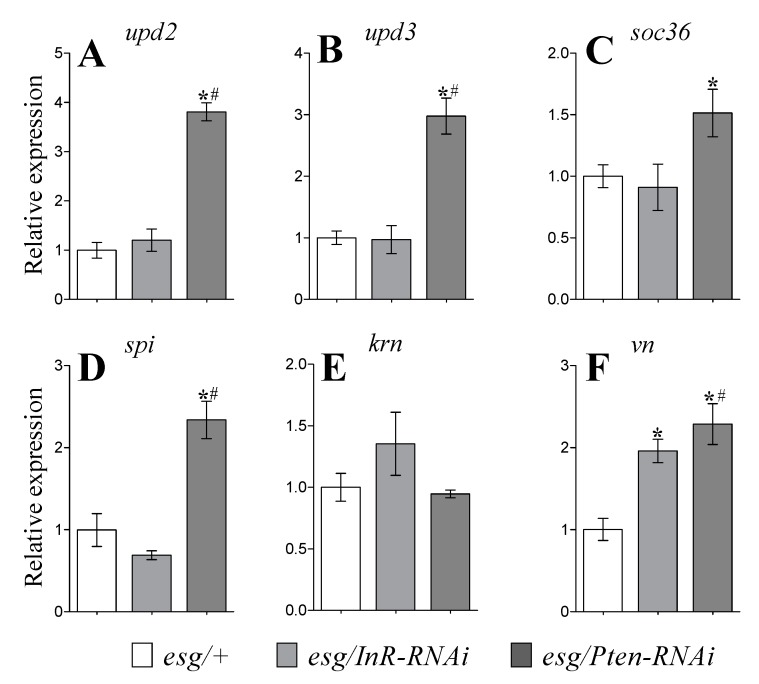
Expression of genes related to JAK–STAT and ligands for EGFR in the gut. Transcriptional levels of cytokine-inducing genes *upd2* (**A**), *upd3* (**B**), and the downstream target *soc36* (**C**) were measured, as well as those for *spi* (**D**), *krn* (**E**), and *vn* (**F**). Flies were tested after six days after expression induction at 29 °C. Data represent mean values for 4 independent measurements (± SEM). Group comparisons were performed using ANOVA followed by Newman–Keuls post hoc test. Asterisks indicate significant difference from the control flies (*esg/+*) (*p* < 0.05); hash signs show significant difference from flies with inhibited IIS (*esg/InR-RNAi*).

**Figure 6 cells-09-00803-f006:**
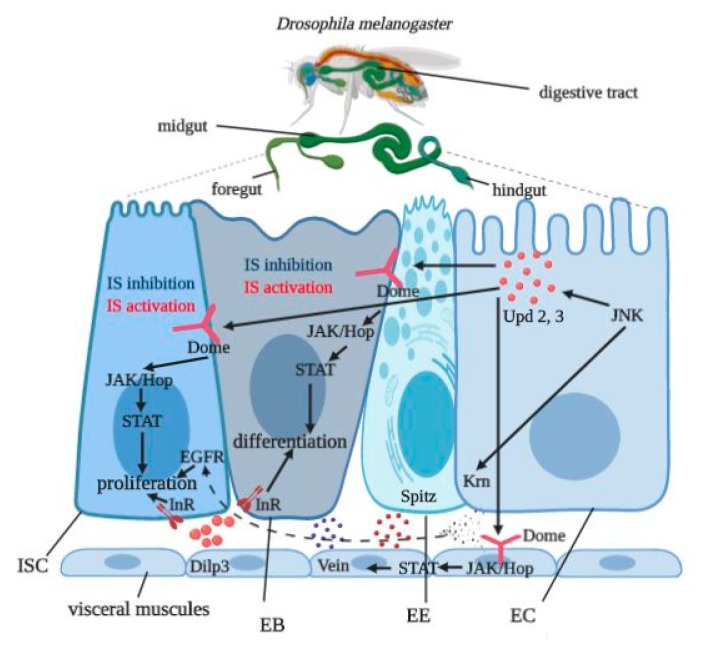
The digestive tract of *Drosophila melanogaster* and niche intestinal stem cell (ISC) of midgut epithelium. IIS was manipulated in ISCs and EBs in the *Drosophila* midgut, and regulated ISC proliferation and EB differentiation. ECs produce signaling ligands Upd2 and Upd3 for JAK–STAT activation in ISCs, EBs, and visceral muscles. JAK–STAT in visceral muscles activates Vein secretion, which together with Spitz and Keren may bind EGFR in ISCs. JAK–STAT, EGFR, and local IIS regulate ISC proliferation and EB differentiation.

**Figure 7 cells-09-00803-f007:**
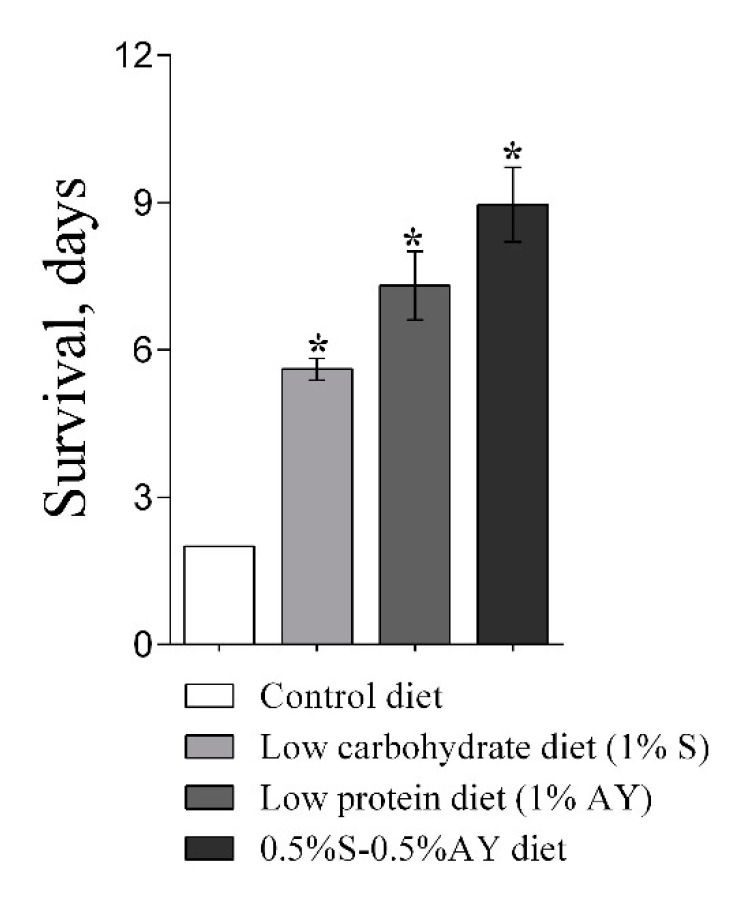
Mean lifespan of flies with inhibition of IIS in ISCs (*esg/InR-RNAi*) under different dietary regimens. Asterisks show a significant difference from the standard dietary condition (5%S + 5%Y) with *p* < 0.05.

**Table 1 cells-09-00803-t001:** Summary of IS modulation in ISCs and EBs on the responses of fly physiological, metabolic, and gene expression parameters.

Parameters	Inhibition	Activation
Lifespan	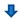	
Starvation resistance	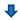	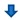
Malnutrition resistance1% sucrose	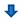	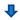
Malnutrition resistance1% autolyzed yeast	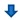	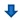
Malnutrition resistance0.5% S and 0.5% AY	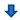	
Feeding, Fecundity	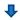	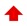
Body glucose, glycogen	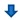	
*dilp2, dilp5, tobi*	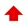	
*dilp6*	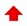	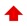
*akh*		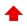
*upd2, upd3, soc36*		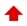
*spi*		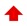
*vn*	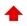	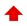

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
