# Peer review of "Insulin Signaling in Intestinal Stem and Progenitor Cells as an Important Determinant of Physiological and Metabolic Traits in Drosophila"

_cells, 2020, doi:10.3390/cells9040803_

Round 1

Reviewer 1 Report

“Insulin Signaling in Intestinal Stem and Progenitor Cells as an Important Determinant of Physiological and Metabolic Traits in Drosophila” by Strilbytska et al., attempts to dissect the metabolic and physiological consequences of altering the insulin signaling (IS) pathway in intestinal stem cells (ISC) and progenitor cells of the Drosophila melanogaster gut. They find that peripheral insulin signaling, and JAK/STAT signaling in the gut may be altered, in addition to whole fly glucose and glycogen levels. Reduced IS appears to negatively regulate feeding, reproduction, lifespan, and starvation resistance.

The experimental plan appears to be quite well thought of: the authors use a number of different assays to examine various parameters. However, I do have significant concerns, as follows:

  1. I am perplexed that control flies maintained on a standard medium (5% sucrose + 5% yeast) have a maximum lifespan of only 30 days? In our own hands (and many other published studies), the median lifespan of wild type or control flies is ~45 days, and maximum lifespan is 70 to 80 days. I wonder if other factors such as stress or dehydration contributed to early mortality of control flies. Can the authors explain this discrepancy?
  2. While insulin signaling is known to affect lifespan, it is remarkable that (incomplete loss-of-function) knockdown of InR in select gut cells could so drastically decrease lifespan in InR-RNAi flies, even on a standard diet, to only 2 days, the same as under complete starvation.
  3. The authors ignore the potential involvement/or interaction with other key longevity pathways such as TOR, particularly since they test fly survival on different diets.
  4. No ‘rescue’ experiments are attempted to strengthen their conclusions regarding the involvement of IS in the reported phenotypes
  5. TAG assays are mentioned in 2.5 of Materials and Methods, but I do not see any TAG measurements in the Results. Indeed, TAG is a very important metabolite, and whole body TAG amounts can have significant effects on starvation resistance and lifespan. The gut has potential for de novo lipogenesis from sugars derived from diet, and can also retain lipids under certain conditions. The lack of TAG measurements in this study is a significant omission that I feel the authors must address.
  6. The authors measure fly glucose, though it is not clear whether they measure glucose or trehalose levels. Trehalose is the dominant sugar in Drosophila hemolymph (fly blood), its concentration 100 - 200 times that of glucose. Also, glucose and trehalose levels do not always track together, meaning a fly can have increased glucose levels (hyperglycemic) but decreased trehalose (hypotrehalosemic). Interestingly, the reverse is not true. In my opinion, the authors should measure both circulating carbohydrate fractions.
  7. While the authors examine gut integrity by the “smurf” assays, they should consider observing dissected guts more closely for changes in cell morphology by microscopy, and for hyperplasia or hyperphagy (they mention that a previous study found that increased upd2 and upd3 expression in gut stem cells caused hyperplasia)
  8. I think in general it would be useful to have a cartoon of the adult fly gut and highlight the cells in which IS was targeted
  9. There is a typo on line 2 of 2.3: Briefly experimental flies were kept in vials supplied with a 5 ul capillary tube.

The study needs to be more comprehensive before it can be considered for publication by Cells.

Reviewer 2 Report

The work described by Strilbytska et al aimed at investigating how modulation of insulin signalling in the midgut ISCs and EBs affects various aspects of physiology and potentially metabolism.

Overall the work relies heavily on three genetic backgrounds, a control with the esg-Gal4 driver, esg>InR RNAi and esg>Pten, and a temperature shift design to switch on the transgenes in adulthood.

Major comments:

This aspect of the design is clever, however I would have liked to see controls that show the Gal80 to be working (e.g. early death when raised at 29).

I am also somewhat concerned that esg-Gal4 (incorrectly called esc here) expresses elsewhere in the adult. As with many of these lines, Gal4 expression is reported in one place (e.g. midgut), but upon inspection, other patterns are also observed. Unless the authors have tested this, and would be willing to include such data, I would suggest including the necessary caveats that there is a possibility that these phenotypes may be at least in part due to other tissues (e.g. ovary).

The other major point I would like openly addressed in the manuscript is the testing of esg>InR RNAi flies for feeding, fecundity and glucose, etc. The methods suggest that these flies were tested for several days (e.g. feeding for four days), yet the survival curve for this genotype suggests that all adults are dead at day two. How do the authors reconcile this?

Lastly, overall there could be a better narrative / lead into the various tests that were performed. At present it reads a little clunky from one experiment to the next. I think the flow would benefit from such changes.

Minor points:

-In the abstract, the phrase 'non-knock-out controls' is mentioned. The tool used is RNAi, so 'knock-down' is the correct term to use.

-In the methods 'four lines of flies' were used, but only three look to be mentioned - please clarify, and also include stock numbers for all.

-Other controls are suggested to give the same result, so aren't shown. I think that these controls should be included. If they truly are the same, these data should strengthen your arguments, but should be subject to peer review.

-The cafe assay method suggests that 'flies were kept in a 5ul capillary'. Please correct as necessary.

-In the malnutrition and starvation assay method, '6/12/6' is mentioned - please clarify what this means.

-In the first part of the results the term 'lifespan regulator' is used. This phrase is not appropriate - 'regulator' implies that it acts to regulate lifespan. IS regulates physiology that impacts lifespan.

-unusual to refer to Figs and panels out of order (Fig1A then Fig1E). Suggest reorder figures. Also refers to first mention of Fig 6.

-define mg/gww in metabolic measurements

-TAG result - I read about it in the methods, but did not see a Fig for these data. Please correct.

-Why was akh measured from whole bodies, not from heads (since it is produced by the CC cells of the brain, unless I am mistaken). Including justification for this would be useful.

-Text at 3.4 says dilp3 is changed, but data (and last line of this results paragraph contradict this)

-Did the authors measure an EGFR target gene (ie. argos)? Please ensure that caution is taken in discussion where an influence on EGFR signalling is suggested - this has not been shown here (but it has for JAK/STAT).

-Smurf data should be included since it is important for the conclusions.

-The formatting for Table 1 did not convert properly - possibly and editorial issue.

Round 2

Reviewer 2 Report

The authors' revised manuscript is significantly improved. However I am not satisfied that the authors have made it clear enough that the esg>InR RNAi flies were measured differently for feeding and fecundity (and presumably metabolism too). I note that the authors have included mention of this for the smurf assay methods (2.6), but this statement does not indicate that measurement was on day one. This should be explicit, and the same statement should be made for feeding and fecundity measurements. I am quite concerned that such an important point be hidden from the reader. For this reason, I would further include this information in the relevant figure legends, so as not to mislead readers. It is critical for the reader to be able to consider that 1) these flies are severely compromised when these measurements were made and 2) because of this, the data are from a greatly reduced sampling period.

Author Response

Response to Reviewer 2 Comments

Point 1: The authors' revised manuscript is significantly improved. However, I am not satisfied that the authors have made it clear enough that the esg>InR RNAi flies were measured differently for feeding and fecundity (and presumably metabolism too). I note that the authors have included mention of this for the smurf assay methods (2.6), but this statement does not indicate that measurement was on day one. This should be explicit, and the same statement should be made for feeding and fecundity measurements. I am quite concerned that such an important point be hidden from the reader. For this reason, I would further include this information in the relevant figure legends, so as not to mislead readers. It is critical for the reader to be able to consider that 1) these flies are severely compromised when these measurements were made and 2) because of this, the data are from a greatly reduced sampling period.

Response 1: We thank a reviewer for these suggestions and added additional information to the main text of the manuscript and figure legends. Introduced changes are marked by red font in the new version of the manuscript. Experimental flies were kept for 6 days at 29°C to induce expression of the UAS constructs. Next, resulted flies were selected and separated for the: lifespan assay, starvation and malnutrition experiments, gut integrity determination, they were also frozen for metabolites content and gene expression determination. After 6 days of expression induction the readout of experimental days starts initially. Since, esc/InR-RNAi flies lived only one experimental day (6 days of expression induction + 1 day), the fecundity and feeding assays were conducted measured only ones. The number of flies for each genotype was the same, that is why the sampling number was the same.

The Smurf assay was repeated every 5 days for up to 30 days (on the 1, 5, 10, 15, 20, 25, 30 day). Since the flies of esg/InR-RNAi genotype died fast gut integrity was only measured ones (only on the 1-st experimental day after six days of expression induction).